# Endurance Training vs. Circuit Resistance Training: Effects on Lipid Profile and Anthropometric/Body Composition Status in Healthy Young Adult Women

**DOI:** 10.3390/ijerph17041222

**Published:** 2020-02-14

**Authors:** Gentiana Beqa Ahmeti, Kemal Idrizovic, Abdulla Elezi, Natasa Zenic, Ljerka Ostojic

**Affiliations:** 1Faculty of Physical Education and Sport, University of Prishtina, 10000 Prishtina, Kosovo; gentiana.beqa@uni-pr.edu (G.B.A.); abdulla.elezi@uni-pr.edu (A.E.); 2Faculty for Sport and Physical Education, University of Montenegro, 81400 Niksic, Montenegro; kemo@t-com.me; 3Faculty of Kinesiology, University of Split, 21000 Split, Croatia; ljerka.ostojic@mef.sum.ba; 4Faculty of Medicine, University of Mostar, 88000 Mostar, Bosnia and Herzegovina

**Keywords:** physical activity, physical exercise, aerobic, circuit weigh training, women, lipid profile

## Abstract

Background: Endurance training (ET) and resistance training (RT) are known to be effective in improving anthropometric/body composition and lipid panel indicators, but there is an evident lack of studies on differential effects of these two forms of physical exercise (PE). This study aimed to evaluate the differential effects of 8-week ET and RT among young adult women. Methods: Participants were women (*n* = 57; age: 23 ± 3 years; initial body height: 165 ± 6 cm; body mass: 66.79 ± 7.23 kg; BMI: 24.37 ± 2.57 kg/m^2^) divided into the ET group (*n* = 20), RT group (*n* = 19), and non-exercising control group (*n* = 18). All participants were tested for cardiovascular risk factors (CRF), including total cholesterol, high density lipoprotein (HDL), low density lipoprotein (LDL), triglycerides, glucose, and anthropometric/body composition (body mass, body mass index, skinfold measures, body fat %) at the beginning and at the end of the study. Over the 8 weeks, the ET group trained three times/week on a treadmill while the RT group participated in equal number of circuit weight training sessions. Both types of training were planned according to participants’ pre-study fitness status. Results: A two-factor analysis of variance for repeated measurements (“group” × “measurement”) revealed significant main effects for “measurement” in CRF. The “group × measurement” interaction was significant for CRF. The post-hoc analysis indicated significant improvements in CRF for RT and ET. No significant differential effects between RT and ET were evidenced. Conclusions: The results of this study evidence improvements of CRF in young adult women as a result of 8-week ET and RT. The lack of differential training-effects may be attributed to the fact that all participants underwent pre-study screening of their fitness status, which resulted in application of accurate training loads.

## 1. Introduction

The “lipid panel” (i.e., lipid profile, lipid test) is a term used to describe levels of lipids in the blood and includes total cholesterol (TC), low-density lipoprotein cholesterol (LDL), high-density lipoprotein cholesterol (HDL), and triglycerides (TG). Because of its importance in evaluating cardiovascular status, the lipid panel is frequently called the “coronary risk panel”, with LDL and TG being strong risk factors for cardiovascular insult [1]. The high-risk profile is correlated with indices of overweight/obesity (independently of other factors such as race, gender, and environment), with higher risk in overweight and obese individuals [2,3,4,5]. Therefore, evaluations of both anthropometric/body composition indices and the lipid panel are valuable tools in assessing risk for cardiovascular problems [6,7].

In the last decade or two, there has been a worldwide increase in obesity and a parallel decrease in cardiovascular fitness. The treatments for this health issue are various such as lifestyle modifications, pharmacological therapies, and surgery [8]. While a low level of physical activity is one of the factors known to be directly related to such trends, studies regularly evidence that increasing physical activity levels (which is often assured by incorporation of physical exercise programs in everyday lifestyle) significantly improves health indices and are advocated as important elements of primary or adjunctive therapy [9]. For better understanding, the difference between “physical activity (PA)” and “physical exercise (PE)” is important [10]. In short, while PA refers to any bodily movement produced and Prevention) vary when it comes to type, intensity, volume, and frequency of PE [11]. This is not surprising knowing that different types of PE have differential effects on specific health indices [2]. 

For example, aerobic exercise (i.e., exercise that involves cardiorespiratory endurance exercises (cyclic activities) such as jogging, walking, dance aerobics, treadmill running, cycling) has been shown to improve the cardiovascular fitness of participants, and such findings are relatively consistent irrespective of participants’ age, sex, and/or fitness status [12,13,14,15,16]. At the same time, resistance training (PE performed in an acyclic manner by exercising a muscle or a muscle group against external resistance) is primarily oriented toward muscular fitness. Consequently, participation in such a form of PE directly improves strength and power capacities [17,18,19]. 

While the benefits of PA in everyday life are relatively well documented, the problem of regularity in PA is crucial. Modern life offers many advantages, but many young adults are facing the problems of extremely busy schedules, including “overload” and or “increased stress levels” [20,21]. Therefore, it is of high importance to find the most appropriate type of PE for the various needs of different subjects. This is a particularly important problem in women. Briefly, there is conclusive body of evidence that “being female” was negatively associated to PA and PE in all age categories (e.g., children, adolescents, and adults) [22] Moreover, it has been hypothesized that discouraging family/social environments could be a significant factor preventing, which is supportive to European Union survey which indicates a consistent decrease in PE in women, particularly those older than 24 years [22,23]. Indeed, women are nowadays frequently employed but at the same time, face greater home duties and parental responsibilities than males. This altogether limits their possibility to achieve regularity of PE, even if they are well informed about the necessity and importance of PE. 

Another factor that directly contributes to PE effectiveness is regularity of exercise (i.e., exercise consistency—EC) [24,25]. The majority of studies that have examined the effects of PE in women have been based on a random sampling experimental approach, where participants were grouped into experimental and control groups by random selection [26,27]. Although undoubtedly methodologically justified, such an experimental approach does not reflect the behavioristic nature of PE. Namely, individuals’ choice of PE type is strongly influenced by self-preferences [28]. Therefore, in order to evaluate the effects of PE that will be transferable to the real world (in other words that will be “ecologically valid”), it is of the upmost importance to identify the possible effects of PE type that were chosen individually by each participant. Meanwhile, there is a limited body of knowledge about the effectiveness of PE when the type of PE is chosen according to participants’ own preferences.

From the literature overview, it is clear that a limited number of studies have examined and compared the effects of resistance training and endurance training in young adult women, especially with regard to important health-related indices such as the lipid panel and anthropometric/body composition indices. Next, information about the effectiveness of resistance training and endurance training in women who deliberately participate in certain PE programs is particularly scarce. Specifically, the initiation and consistency in PE is strongly correlated to different psychological and behavioral characteristics (including self-motivation, personal characteristics, environmental specifics), and it is hard to expect that individuals will partake in exercise program they do not prefer [29,30]. This study aimed to evaluate the differential effects of 8 weeks of endurance training or resistance training on the lipid profile and anthropometric/body composition indices of young adult healthy women who self-decided to participate in one of the studied PE programs. Initially, we hypothesized that (i) endurance and resistance training will induce positive changes in studied variables; (ii) endurance training will reveal better results, with regard to positive changes in the lipid profile of the participants, than resistance training; and (iii) resistance training will reveal more superior results with regard to positive changes in anthropometric/body composition indices than endurance training.

## 2. Materials and Methods

### 2.1. Participants

The participants in this study were young healthy adult women (*n* = 57; age: 23 ± 3 years; initial body height: 165 ± 6 cm; body mass: 66.79 ± 7.23 kg; BMI: 24.37 ± 2.57 kg/m^2^). They were all members of one fitness center in Pristina, Kosovo and participated in this study between February and June 2018. None of them had any training history and they were mostly University students. All participants deliberately chose one of the examined PE programs; therefore, the total sample was divided into the resistance training group (*n* = 19), endurance training group (*n* = 20), and control (non-exercising) group (*n* = 18). The control group consisted of those women who were interested in being involved in some of the studied PE programs, but at the moment of their first visit to fitness center their preferred PE program was unavailable (due to limited space/equipment and/or currently large number of participants). On the basis of the pre-testing lipid-profile results (please see later for details on testing), the above optimal level of TC was evidenced in 25% of participants (5% with high levels of TC), 23% of participants had above optimal level of LDL (6% with high levels of LDL), and <10% of participants had above optimal levels of TC [31]. A total of 32% of participants were overweight/obese (BMI > 25 kg/m^2^). All participants were informed of the benefits and risks of participating in the study and they signed informed consent forms for their participation. They were informed that they could leave the program at any time. The study was conducted in accordance with the Helsinki declaration and was approved by the Ethical Board of the University of Split, Faculty of Kinesiology, Split, Croatia (EBO: 2141-6775-234).

### 2.2. Testing and Variables

All the participants were tested at two time points: initial testing (pre-testing) and at the end of the two-month period (post-testing). Variables included anthropometric/body composition indices, lipid panel, and nutritional intake. Anthropometrics were tested in an accredited medical laboratory where blood samples were also drawn (Biohem Laboratory, Gjakovë, Kosovo).

Anthropometric/body composition variables were measured with a Seca stadiometer and scale (Seca, Birmingham, UK) and skinfold caliper (Holtain, London, UK), and measurements included body height, body mass, triceps skinfold, thigh skinfold, and suprailiac skinfold. An experienced technician measured all anthropometric variables in the morning, prior to blood sampling. Later, the body mass index (BMI) was calculated by dividing participants’ body mass (kg) by their squared body height (m). The body fat percentage was calculated by the Jackson–Pollock formula for body density, and Siri equation for body fat percentage (BF%) [32]:Body density = 1.0994921 − (0.0009929 × [triceps skinfold + thigh skinfold + suprailiac skinfold]) + (0.0000023 × [triceps skinfold + thigh skinfold + suprailiac skinfold] 2) − (0.0001392 × age),BF% = (4.95/body density − 4.5) × 100.

Blood samples were taken after anthropometric measurements, after overnight fasting, from the antecubital vein. Samples were collected in BD Vacutainer^®^ SSTII Advance vacuum tubes (BD, Plymouth, UK) and centrifuged at 3500 rpm for 10 min (Centrifugal Hettich, Tuttlingen, Germany). Plasma glucose (GLU), total cholesterol (TC), high-density lipoprotein (HDL), and triglycerides (TGs) were analyzed with the COBAS Integra 400+ analyzer (Roche Diagnostics International Ltd., Rotkreuz, Switzerland). The Friedewald equation was used for estimating concentrations of LDL [33].

Nutritional intake was controlled at study baseline and at the end of the study. Participants were asked to fill out food diaries over 3 days at the beginning and at the end of the study (see Figure 1). Later, energy intake was analyzed using nutritional tables and software [34]. The study protocol is presented in Figure 1.

### 2.3. Physical Exercise Programs

Both PE programs lasted two months (8 weeks), with a training frequency of 3 sessions per week (Monday, Wednesday, Friday). Initially, the authors discussed the duration of the experiment and agreed that the majority of females participated in 2-month (8-week) training, and after that moment, significant drop-out regularly occurs. While the authors were interested to investigate the effects of real-life training, the duration of 8-weeks was identified as being appropriate. Altogether, participants performed a maximum of 24 sessions, but herein, we included all the participants who were present for a minimum of 21 training sessions. All the sessions took place from 16:00 to 21:00. In this study, training programs were not matched on the basis of energy expenditure but in terms of duration. Therefore, both training programs lasted for 45–60 min and progressed over the course of the study (i.e., from 45 min from the beginning, up to 70 min at the study end).

Endurance training consisted of aerobic endurance treadmill walking/running exercises performed on Nova 450 equipment (Nova Sport, Istanbul, Turkey). This training equipment allows inclination of the surface from 0% and 15%, speed range from 1 to 20 km/h, and an effective running surface of 56 × 150 cm. Before the pre-testing first training session, each participant was tested by the Conconi test, a noninvasive method for assessing individual anaerobic thresholds by the heart rate threshold (i.e., deflection point from the linear relationship between work load and heart rate) [35]. The heart rate threshold has been shown to be correlated significantly with the anaerobic threshold in healthy individuals and later, it was shown to be applicable for evaluating the anaerobic threshold even in participants with certain health problems [36]. This protocol allowed us to determine the individual heart rate threshold for each participant, which was later used as an indicator of maximal individual training load. Participants exercised in a range of 5–30 beats bellow the anaerobic threshold, which corresponds to 120–155 beats/min (60%–80% of maximal heart rate) on average. The Conconi test was applied every two weeks in order to track the progress and to eventually redefine training loads. Throughout the course of the study, participants involved in endurance training performed treadmill walk/run exercises while heart rate values were monitored by a heart rate monitor. During the endurance exercise, participants performed different forms of endurance-based training protocols (i.e., continuous, interval (starting from 2 min exercise + 1 min rest in first week up to 4 min exercise + 1 min rest last week), Fartlek) but heart rate values were constantly kept to at least 5 bpm below the identified threshold. In general, participants were instructed to perform one form of endurance-based training protocol throughout the one week and then to use another one (i.e., continuous training first week, Fartlek, the next week, and so on), but this was not mandatory, and participants frequently changed training forms within the single week. Generally, all participants participated in different forms of endurance exercise, meaning that none of the participants performed only one modality of exercise over the study course. Training sessions were programmed and controlled according to the individual needs of each participant and intensity was modified by inclination and/or speed of the treadmill carpet.

Resistance training was organized and performed as circuit weight training using handheld weights, weight machines (Technogym, Cesena, Italy) and participants’ own body weight. Prior to the study, each participant performed two training sessions in order to accommodate herself with the training equipment and to familiarize herself with proper techniques and execution of exercise. Moreover, throughout these “familiarization sessions”, the training instructor noted the appropriate weights for each exercise and each participant. This allowed us to make the chart where weights (resistances) were evidenced and later used in the resistance training. Every two weeks (3 times over the study course), participants were invited to participate in an additional individual training session where first author of the study tested them on exercises used in circuit-weight-training in order to re-define individual training loads for the upcoming period of two weeks. The general structure of the applied circuit weight training is presented in Table 1. Resistance exercise was performed in groups of 4–6 participants but each participant followed individual guidelines (i.e., specific technique, weight) that were posted at each exercise station.

### 2.4. Statistics

Kolmogorov–Smirnov tests were used to evaluate the normality of the distributions for all variables, and descriptive statistics included means and standard deviations. Levene’s test was used to assess the homoscedasticity of variables.

A two-way analysis of variance (ANOVA) with repeated measures (Time (pre- and post-training) × Group (resistance training conditioning, endurance training, control)) was used to determine the effects of PE on the studied variables, with Scheffe’s post-hoc analyses. To evaluate the effect sizes, the partial eta squared values (*η*^2^) were also reported (small effect size [ES]: >0.02; medium ES: >0.13; large ES: >0.26) [37,38]. A significance level of *p* < 0.05 was applied and Statistica 13 (TIBCO Software Inc. Palo Alto, CA, USA) was used.

## 3. Results

Descriptive statistics for the studied anthropometric/body composition-, and lipid panel-variables are presented in Table 2.

Significant ANOVA main effects for “Group” were evidenced for thigh skinfold (medium ES), BF% (medium ES), and HDL (medium ES). The main effects for “Measurement” were significant for all studied variables, namely body mass (large ES), BMI (large ES), triceps skinfold (large ES), thigh skinfold (large ES), suprailiac skinfold (large ES), BF% (large ES), TC (large ES), HDL (large ES), LDL (large ES), TG (large ES), and PG (large ES) (Table 3).

The “Measurement × Group” interaction effects reached statistical significance for all studied anthropometric/body composition and lipid-panel-variables, namely body mass (large ES), BMI (large ES), triceps skinfold (large ES), thigh skinfold (large ES), suprailiac skinfold (large ES), BF% (large ES), TC (large ES), HDL (large ES), LDL (large ES), TG (large ES), and PG (large ES) (Table 3).

The post-hoc Scheffe test revealed significant within-group differences (e.g., pre- to post-testing differences) for both training-groups in body mass, BMI, triceps skinfold, thigh skinfold, subscapular skinfold, BF%, TC, HDL, LDL, PG, and TG (significant decrease in numerical results for all variables in training-groups). For the controls, significant post-hoc within-group differences were found for TG (significant decrease of TG over the course of the study). Between-group post-hoc analysis showed higher BMI for resistance- than for endurance-training-group in pre-testing results. When post-hoc analyses were calculated for post-testing results, significant between-group differences were found for triceps skinfold, BF%, HDL, TG, and PG (all larger in control-group than in both training groups); TC and LDL (both larger in control-group than in endurance-training). No significant post-hoc differences were found between resistance-training and endurance-training group in post-testing.

Caloric intake and nutrient intake (in kcal), did not differ between groups nor among groups (Figure 2).

## 4. Discussion

This study aimed to evaluate the differential effects of endurance training and resistance training on the lipid panel and anthropometric indices of adult healthy women. There were the two most important findings. First, both training modalities induced positive changes in all studied variables. Therefore, the first study hypothesis may be accepted. Secondly, there was no evidence of differential effects of observed training modalities on studied variables. As a result, the second and third study hypotheses were not confirmed.

### 4.1. Positive Effects of Endurance and Resistance Training on Anthropometric/Body Composition Indices

Endurance training induced significant positive effects on anthropometric/body composition indices. Such results were expected and similar improvements were reported in other studies where different forms of endurance (aerobic) training were studied in women. For example, a recent Spanish study with sedentary females investigated the effects of a 16-week Zumba aerobic dance program and revealed positive effects on BMI and body fat percentage [27]. The Lithuanian study where the authors examined the effects of a two-month endurance exercise program performed by indoor cycling, induced a significant reduction of body mass of 1 kg in young health females (in our study, 4 kg), but in this study’s authors examined participants with initially lower BMI and BF% than we have observed herein (BMI: 23 and 24 kg/m^2^, BF: 31 and 35% in Lithuanian and our study, respectively), which naturally resulted in smaller numerical changes in body mass [39]. Positive results of endurance training on different indices of body build have been reported in other studies with young health females as well [40]. On the other hand, it is important to note that the duration of our study was evidently shorter (i.e., 8 weeks) than the durations for the majority of previous studies, which regularly lasted 3 months or more [27,40]. In explaining the significant effects on body composition indices in our study, irrespective of the relatively short duration, the two most important issues should be highlighted and discussed: (i) controlled intensity and (ii) type of endurance exercise.

It is well known that moderate aerobic exercise for > 150 or 200–300 min per week can significantly reduce body mass (weight), even without controlling diet [41]. Proper intensity of endurance exercise is one of the most important factors of exercise effectiveness. Specifically, when exercise intensities differ, exercise expenditure is not the only factor responsible for reducing body mass [42]. High-intensity training may effectively reduce body fat and when energy expenditure is held equal (as it was the case in our endurance group), high-intensity exercise is more beneficial for improving body composition than low-intensity exercise [43]. Prior to the study, each participant involved in endurance PE underwent initial screening where their anaerobic threshold was evidenced (see Methods for details). This allowed us to determine the intensity level (using the treadmill belt speed and heart rate) at which participants would most effectively exercise.

Secondly, it must not be ignored that treadmill exercise was highly convenient for the purposes of this investigation. The treadmill equipment allows inclination of the platform which increases the exercise intensity without increasing the step frequency. As a result, our participants were able to exercise while “walking uphill”. This minimized the stress on the locomotor system [44,45] and allowed convenient and effective endurance exercise. Together with continuous control of exercise intensity by heart rate monitoring, treadmill use almost certainly contributed to the significant effects seen in endurance training, irrespective of the relative short duration of the training program (i.e., 8 weeks).

Resistance training is a popular and convenient form of exercise and its positive effects on anthropometric/body composition status in women of different ages and with various health conditions have already been documented [46,47]. In our study, we applied organized circuit weight training as a form of resistance training. This form of training was developed in the early 1950s and the term “circuit” refers to a series of specific and organized exercises arranged in a sequence [48,49]. Selected exercises can involve different forms of “resistance exercises”, including elastic resistance, weight machines, handheld weights, etc. Participants perform one exercise in a pre-defined order and a number of repetitions (or in a pre-defined time) and after a relatively strict pause, they move to another exercise (for details on circuit resistance training applied in this study, please see Methods). Although originally designed with the main intention to improve strength and cardiorespiratory function, changes in body build and body composition indices are also regularly achieved because of the relatively high metabolic costs of this form of training [48]. However, it must be noted that high-intensity (heavy load) training was previously not effective when it came to changes in anthropometric/body composition parameters in the low-trained women [50].

The effects of circuit resistance training in our study were positive, with significant reductions of body mass, BMI, and BF%. Therefore, the changes achieved as a result of circuit weight-training are comparable to those reported in other research with female participants. In brief, Brazilian authors applied the same type of resistance training in sedentary women (33–45 years of age) who had a similar body composition status as our participants, and evidenced significant decreases in BF% (from 37% to 31) [51]. Further, similar results were reported in an Australian study with obese women [52], while an Iranian team confirmed the effects of 8-week circuit weight training on body weight reduction in postmenopausal women [53]. On the other hand, the positive effects of 8-week circuit resistance training on body composition were not confirmed in a US study on premenopausal women, although numerical values of BMI and BF indices showed promising trends [52]. However, it must be noted that participants in the latter study trained two times a week, which probably explains the lack of effectiveness due to the lower energy expenditure (metabolic demands).

### 4.2. Positive Effects of Endurance and Resistance Training on Plasma Glucose and Lipid Profile

The significant reduction of the PG levels in our participants was expected because: (i) carbohydrates (e.g., PG and muscle glycogen) are one of the main energy sources during PA, and (ii) PA increases insulin sensitivity, which reduces the PG concentration in trained subjects [54,55,56]. On the other hand, the differential effects of studied PE-programs were also possible. In brief, the PG utilization increases with the intensity of PE because of the increase in glucose utilization by muscles. Throughout the PE, the utilization is affected (e.g., increased) by (i) the intensity of the activation of each muscle unit and (ii) the increase in the number of active muscle units, and/or (iii) both [57]. Although we cannot ignore the fact that both of these mechanisms are more prevalent in resistance training (i.e., resistance training increase the intensity and number of activated muscle unites to a greater extent than endurance-training), plasma glucose utilization also increases with the duration of exercise. It therefore may at least partially compensate the theoretically expected superior effects of resistance-training in our study.

The association between high cholesterol levels and ischemic heart disease has been known for almost 20 years. Results of studies that aimed to reduce cholesterol levels showed reduced risk of cardiovascular diseases as a consequence of lower cholesterol levels [58]. It is important to note that the prevalence of high cholesterol has decreased globally over the last 20 years but a significant proportion of this decrease may be attributed to the use of cholesterol-lowering substances. For example, it is estimated that usage of cholesterol-lowering substances in US citizens has increased more than 400% starting from the early 1990s [59]. Therefore, our results of the significant effects of both observed PE modalities on lowering cholesterol levels are encouraging, especially knowing that our participants did not use cholesterol-lowering substances, were not involved in a specific nutrition program, and/or, to the best of our knowledge, did not change their caloric intake over the course of the study.

Our results show positive effects of aerobic endurance exercise on changes in lipid profile, and this is generally consistent with previous reports where authors examined longer training programs. For example, a 6-month study of Dunn et al., who examined a mixed sample of healthy, sedentary men and women, showed promising results, with significantly positive changes in total cholesterol, LDL, and the total/HDL ratio [60]. One of the rare studies exclusively examining females evaluated the effects of 16 weeks of endurance exercise and a significant increase in HDL and decrease in the concentration of triglycerides were reported [40]. In a recent study with healthy young females (25–30 years of age), Kyrolainen et al. evidenced changes in lipid panel indicators as a result of 9-week endurance training [11]. More specifically, TG changes were not significant (1.17 ± 0.34 and 1.01 ± 0.29 mmol/L), TC was significantly reduced (5.0 ± 0.05 mmol/L and 4.4 ± 0.7 mmol/L), while HDL levels significantly increased (1.37 ± 0.29 and 1.51 ± 0.27 mmol/L, before and after study period, respectively) [11]. The somewhat better improvements in our study may be explained by the fact that our participants performed treadmill-endurance training, while Kyrolainen et al. applied indoor cycling as a type of endurance training. Knowing the differences in overall energy expenditure between these two forms of exercising (i.e., treadmill walking/running engage larger percentage of the musculature than cycling), the somewhat better effects in our study are understandable [61]. However, despite some differences, our results support previous findings which evidenced the positive effects of exercise in healthy young females.

Interestingly, in our study, the resistance training program induced similar positive changes in lipid profile as endurance training. Although we expected that endurance training would have better effects on lipid profile variables than resistance-training, reports which highlight significant improvements in cardiovascular risk factors as a result of resistance-based PE are not rare. For example, Prabhakaran et al. studied premenopausal women and reported significant decreases in total cholesterol (from 4.6 to 4.26 mmol/L) and LDL cholesterol (from 2.99 to 2.57 mmol/L) as a result of 14-week resistance training [62]. Similarly, in a study with healthy males, high-resistance training and moderate-intensity resistance PE performed over 6 weeks were equally effective with regard to reductions of LDL, total cholesterol, and the total cholesterol: HDL ratio [63]. Finally, circuit weight training, very similar to the one applied in our study, resulted in significant reductions in total cholesterol and triglycerides in a study of obese women [26], altogether indicating clear benefits of resistance training on improving the lipid profile of female participants.

### 4.3. Study Limitations and Strengths

This study has several limitations that must be mentioned. First, we compared three-day logs obtained at the beginning and at the end of the study, but did not continuously monitor diets, and this investigation did not include strict control of the participants’ diet. Next, we did not observe daily physical activity of women involved in the investigation, which may also contribute to changes in studied variables. Moreover, the study was not based on total randomization, but participants were involved in PE types according to their preferences. Although this may be observed as a study limitation, the authors believe that it may also contribute to the ecological validity of the investigation. Namely, self-motivation for PE is among the most important factors of initiation, and consistency in PE [24,29,30]. Therefore, since participants autonomously selected the type of exercise they will participate in, our results may actually reflect the effects that may be expected in “real-life”.

This is one of the rare studies where programmed and controlled endurance and circuit resistance training were observed as methods of PE in young adult women. This allowed us to clearly identify the effects of PE itself, irrespective of eventual confounding effects of individual characteristics (technical knowledge of exercise execution, monitoring of exercise intensity, etc.). In addition, participants were women who self-decided to start physical exercise and who deliberately chose the PE-type; therefore, the results obtained may be observed as plausible for the general population of young healthy women.

## 5. Conclusions

Our results confirm the positive effects of endurance training, performed on treadmill, and circuit resistance training on lipid panel and anthropometric/body composition indices in adult healthy women. Moreover, no differential effects between the applied PE programs were evidenced. Since the applied PE programs lasted 8 weeks, it may be concluded that both types of exercise were equally effective with regard to changes in the observed parameters of cardiovascular risk. However, it must be mentioned that both PE programs had several specifics that probably influenced their effectiveness despite the differences in exercise character (e.g., resistance vs. endurance training).

Circuit resistance training included exercises done with own body weight, weight machines and handheld weights. Since all participants were tested on their strength capacity prior to the study, individualized training programs were developed and applied throughout the study course. In addition, every two weeks, additional screening of the strength capacities was done, which allowed us to redesign the volume and intensity of training. Endurance training consisted of treadmill exercising, and prior to study, all participants were tested for their initial endurance capacity. The result of each participant was then recorded and used in the training design, which, together with training load monitoring, resulted in appropriate and precise workloads in each training session. Altogether, this design almost certainly influenced the efficacy of both PE modalities.

This study involved apparently healthy female participants 32% of whom were overweight/obese and 25% of participants who had above-optimal levels of lipid-panel indicators. Therefore, the necessity of the study may be questioned. However, because of the alarming trends of increase in cardiovascular-risks, it is important to determine the eventual positive effects of PE even in low-risk populations. Namely, this study demonstrates that exposure to individually preferred PE can induce significant health benefits and can therefore prevent the occurrence of various cardiovascular risks. This information can be used in targeted and preventive public health efforts.

In future studies, the observed forms of PE here should be evaluated with regard to their effects on other important fitness-related variables. In doing so, special attention should be placed on strength, endurance, and flexibility components. In addition, similar studies for other types of PE are warranted.

## Figures and Tables

**Figure 1 ijerph-17-01222-f001:**
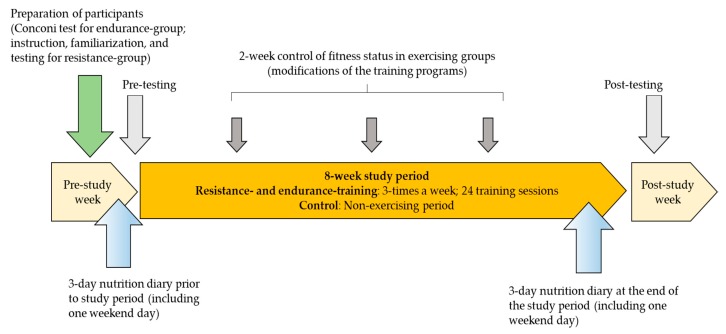
Study protocol.

**Figure 2 ijerph-17-01222-f002:**
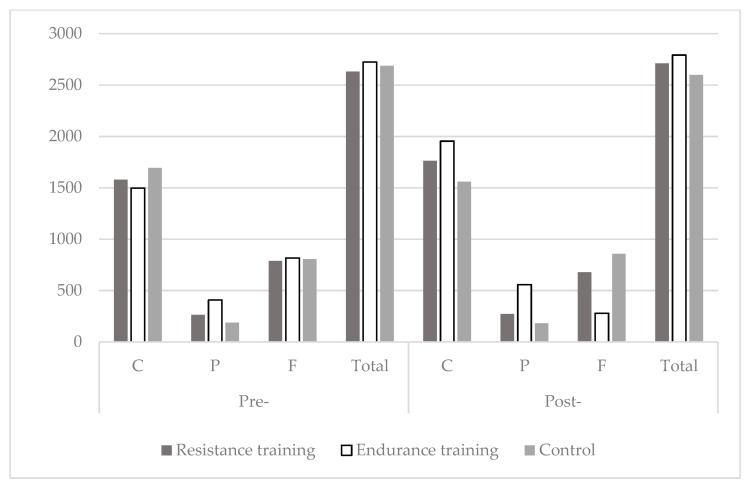
Caloric intake (Total, Carbohydrates-C, Proteins-P, Fats-F; all in kcal) for the resistance-training-, endurance-training-, and control-group at the beginning (Pre-), and at the end of the study (Post-).

**Table 1 ijerph-17-01222-t001:** Characteristics of the applied resistance training (circuit-weight-training).

Exercise	Equipment used	Week 1	Week 2	Week 3	Week 4	Week 5	Week 6	Week 7	Week 8
Abdominal curls	OBW	x	x	x				x	
Abdominal curls (added weights)	OBW + FW				xx	xx	xx		xx
Leg raises	OBW							x	
Hyperextension bench	OBW	x	x	x	xx	xx	xx	x	x
Knee extension	WM	x				x			
Hamstring curl	WM	x	x	x	x	x	x	x	
Lunges	OBW		x			x	x		
Lunges (added weights)	OBW + FW							x	x
Squats	OBW			x	x		x		
Squats (added weights)	OBW + FW							x	x
Legg adductions	WM	x	x	x		x			x
Legg abductions	WM	x	x	x			x		x
Latt pulldowns	WM	x	x				x		x
Rowing torso	WM			x	x	x		x	
Biceps curls		x	x	x	x			x	
Triceps extension	WM	x	x	x					
Butterfly bench	WM	x	x					x	
Bench press	WM & FW			x	x	x	x		x
Inclined bench press	WM & FW								
Overhead press	WM & FW				x			x	x
Number of circuits per training session		2	2	3	3	2	3	3	2
Time of work per set (in seconds)		30	30	25	30	30	30	30	30
Rest between sets (in seconds)		30	30	35	30	30	30	30	30
Rest between circuits (in minutes)		5	5	5	5	5	5	5	5
Warm-up (in minutes)		10	10	10	10	10	10	10	10
Cool-down (in minutes)		10	10	10	10	10	10	10	10

LEGEND: WM-weight machines, FW-free weights, OBW-own body weight, “x” indicates the number of sets for each exercise in one circuit.

**Table 2 ijerph-17-01222-t002:** Descriptive statistics of the variables in Pre- and Post-testing for study groups (data are presented as Means ± Standard Deviations; * presents significant pre–post-testing differences in each group).

Variables	Resistance-Training (*n* = 19)		Endurance-Training (*n* = 20)		Control (*n* = 18)	
	Pre-	Post-	Pre-	Post-	Pre-	Post-
BM (kg)	66.53 ± 6.51	62.4 ± 6.58 *	67.35 ± 8.39	62.38 ± 8.56 *	64.89 ± 7.05	64.06 ± 7.78
BMI (kg/m^2^)	24.02 ± 2.39	22.82 ± 2.39 *	25.11 ± 2.69	23.09 ± 2.79 *	23.55 ± 2.32	23.99 ± 2.32
Tr_SF (mm)	24.71 ± 6.3	20.56 ± 3.56 *	26.78 ± 6.58	21.39 ± 4.03 *	23.09 ± 9.84	23.47 ± 9.56
Th_SF (mm)	23.68 ± 9.99	21.06 ± 4.88 *	25.74 ± 7.52	23.87 ± 5.46 *	22.56 ± 5.2	22.58 ± 8.83
SI_SF (mm)	18.27 ± 7.04	15.24 ± 3.02 *	20.57 ± 4.41	17.7 ± 3.7 *	17.76 ± 5.68	17.56 ± 6.4
BF (%)	34.92 ± 5.79	29.89 ± 4.41 *	36.43 ± 5.43	30.14 ± 4.04 *	33.99 ± 5.92	34.05 ± 6.11
TC (mmol/L)	4.57 ± 0.6	3.98 ± 0.55 *	4.94 ± 0.56	3.77 ± 0.47 *	4.63 ± 0.59	4.51 ± 0.57
HDL (mmol/L)	1.63 ± 0.25	1.09 ± 0.16 *	1.74 ± 0.27	1.25 ± 0.18*	1.57 ± 0.21	1.59 ± 0.26
LDL (mmol/L)	3.42 ± 0.48	2.5 8± 0.47 *	3.36 ± 0.43	2.21 ± 0.46 *	3.14 ± 0.62	2.96 ± 0.51
TG (mmol/L)	1.15 ± 0.15	0.6 ± 0.13 *	1.18 ± 0.23	0.72 ± 0.16 *	1.19 ± 0.22	1.01 ± 0.23 *
PG (mmol/L)	5.19 ± 0.55	4.06 ± 0.42 *	4.89 ± 0.54	4.25 ± 0.6 *	5.04 ± 0.51	4.92 ± 0.36

LEGEND: BM-body mass, BMI-body mass index, Tr_SF-triceps skinfold, Th_SF-thigh skinfold, SI_SF-suprailiac skinfold, BF-body fat, TC-total cholesterol, HDL-high density lipoprotein, LDL-low density lipoprotein, TG-triglycerides, PG-plasma glucose.

**Table 3 ijerph-17-01222-t003:** Results of the analysis of variance for main effects (Group and Measurement), and interaction (Group × Measurement) with effect size values (*η*^2^).

Variables	Main Effects						Interaction		
	Group			Measurement			Group × Measurement		
	F-Test	*p*	*η* ^2^	F-Test	*p*	*η* ^2^	F-Test	*p*	*η* ^2^
BM (kg)	0.1	0.88	0.01	327.1	0.001	0.87	128	0.001	0.72
BMI (kg/m^2^)	0.63	0.53	0.02	143.97	0.001	0.73	60.77	0.001	0.69
Tr_SF (mm)	0.93	0.39	0.03	73.73	0.001	0.58	21.36	0.001	0.44
Th_SF (mm)	6.31	0.01	0.19	146.63	0.001	0.73	36.25	0.001	0.57
SI_SF (mm)	1.75	0.18	0.06	39.1	0.001	0.42	8.73	0.001	0.25
BF (%)	3.98	0.03	0.13	183.91	0.001	0.77	46.45	0.001	0.63
TC (mmol/L)	1.75	0.18	0.06	84.62	0.001	0.61	19.45	0.001	0.42
HDL (mmol/L)	5.63	0.01	0.17	139.33	0.001	0.72	38.17	0.001	0.59
LDL (mmol/L)	1.97	0.15	0.07	173.77	0.001	0.77	26.66	0.001	0.49
TG (mmol/L)	7.37	0.01	0.21	464.25	0.001	0.89	35.41	0.001	0.57
PG (mmol/L)	4.54	0.02	0.14	98.08	0.001	0.65	20.65	0.001	0.43

LEGEND: BM-body mass, BMI-body mass index, Tr_SF-triceps skinfold, Th_SF-thigh skinfold, SI_SF-suprailiac skinfold, BF-body fat, TC-total cholesterol, HDL-high density lipoprotein, LDL-low density lipoprotein, TG-triglycerides, PG-plasma glucose.

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
