# Peer review of "Endurance Training vs. Circuit Resistance Training: Effects on Lipid Profile and Anthropometric/Body Composition Status in Healthy Young Adult Women"

_ijerph, 2020, doi:10.3390/ijerph17041222_

Round 1

Reviewer 1 Report

I would like to thank the authors for carrying out their research and constructing their submitted manuscript. The study investigates the effect of endurance training and a circuit-based resistance training upon body composition and blood lipid panels, in comparison to a non-exercising control group. I commend the authors for investigating these effects within a female population in an ever-growing body of research on the male population. Further, I do appreciate the authors rationale for utilising a ‘real life’ approach to their design, i.e. participants choosing their exercise modality.  Overall, the manuscript is constructed to an acceptable standard of English, with only a few grammatical errors that need proof-reading. I have provided my comments, suggestions and questions below, which I believe will help strengthen this manuscript and provide a greater level of clarity and understanding for the reader. 

Introduction:

Lines 36 onwards, I would recommend removing “measure of x in all the lipoprotein particles” and simply leave the abbreviation e.g. “low density lipoprotein cholesterol (LDL)”.

Lines 40-41: I feel this sentence is missing a reference as this is quite a significant point and inherently forms a basis for your rationale.

Line 76-78: While this may be a valid comment within the specific population study, I am not convinced how this rationale transfers to a general population. I do, however, agree to the fact there is a significant need for more research to be carried out in females (both young and old); due to the plethora of exercise studies utilising males. However, the importance of PE should transfer to all individuals. As far as I am aware, you haven’t specifically targeted this population (i.e. young mothers) within the study recruitment? I would suggest rewording this statement to reflect the above points.

Methods:

Participant information: Require a statement in regard to the declaration of Helsinki.

Lines 130/131: Just place units within the brackets, do not require “in”

In regard to the endurance exercise protocol, as far as I understand, you utilise the Conconi test to assess heart rate threshold and have individuals train at approximately 5bpm under this. However, I noticed that you state: “participants performed different forms of endurance-based training protocols (i.e., continuous, interval, Fartlek)”. I wondered how this training modality was chosen, i.e. did they complete one session of each a week – or alternatively did individuals choose their own session each time. In which case, is it possible that individuals completed only interval training, or only continuous? Further, what were the specifics of these training modalities – i.e. on interval training sessions, was this a set protocol for all individuals, x seconds/minutes on, y seconds/minutes rest?

Further, I wondered as to how you managed progression. You correctly reassess their threshold every 2 weeks and adjust their training. However, is the training adjusted via speed, incline, both – or again is this on an individual basis? I do notice that in the discussion you mention “This allowed us to determine the intensity level (using the treadmill belt speed and heart rate) at which participants would most effectively exercise”. However, this is required within the methods and an explanation provided.  

I appreciate the aim of the study was to keep the set up as ‘real life’ and applicable as possible and you do mention that training sessions were programmed and controlled according to individual needs of each participant. However, the lack of specific detail does raise the question of whether there is too much variability. I would advise, where possible to be more specific so that the reader can understand exactly what occurred and what training stimulus was provided.

The resistance-based circuit training includes a large amount of exercises, targeting the whole body which is pleasing to see. While I appreciate you cannot always dictate the number of reps on each exercise (due to the nature of the modality), I did wonder whether the intensity of training was set for those which used machine or free weights. Naturally this would still be different between individuals but ensure training was relative between them. Similarly to my previous comments, perhaps a little more clarity on the specifics of the training would help the reader. 

Results:

Table 2: If post hoc statistics have been completed on the pre-post data, then I would suggest highlighting the significant differences within this table. It will very quickly allow the ready to see exactly what has changed with each training modality.

Lines 227-234: Perhaps these comparisons could be displayed in a simpler way (similar to that in lines 219-222), e.g. significant differences were found between both training modalities and control for triceps skinfold, BF%, TC, HDL, LDL, TG and PG (all greater in control).

In regard to the calorie intake, it is naturally an important result that you do not see any differences between groups nor between time points (in any group). However, I wondered if it may also be helpful to observe the macronutrient distribution. Whilst the calorie intake may be the same, it is naturally possible that individuals taking part in exercise 3 days a week may be inclined to substitute unhealthy (i.e. higher fat) foods for healthier alternatives. Further, this may be of particular interest in consideration of the control group reducing their TG despite not partaking in any exercise or altering their current activity. 

Discussion/conclusions:

Overall, a nicely constructed discussion with justified conclusions reached. You provide good context for the results you have found and discuss any anomalies between studies. I also believe you have adequately tackled the limitations of the study design and provide further rationale for enabling self-choice for participants.

Author Response

I would like to thank the authors for carrying out their research and constructing their submitted manuscript. The study investigates the effect of endurance training and a circuit-based resistance training upon body composition and blood lipid panels, in comparison to a non-exercising control group. I commend the authors for investigating these effects within a female population in an ever-growing body of research on the male population. Further, I do appreciate the authors rationale for utilising a ‘real life’ approach to their design, i.e. participants choosing their exercise modality.  Overall, the manuscript is constructed to an acceptable standard of English, with only a few grammatical errors that need proof-reading. I have provided my comments, suggestions and questions below, which I believe will help strengthen this manuscript and provide a greater level of clarity and understanding for the reader. 

RESPONSE: Thank you for recognizing the quality and potential of our work. We have tried to follow all your comments and amended the manuscript accordingly. Please see bellow how we responded and where to find amendments. Staying at your disposal.

Introduction:

Lines 36 onwards, I would recommend removing “measure of x in all the lipoprotein particles” and simply leave the abbreviation e.g. “low density lipoprotein cholesterol (LDL)”.

RESPONSE: Amended accordingly. Text now reads: “The “lipid panel” (i.e., lipid profile, lipid test) is a term used to describe levels of lipids in the blood and includes total cholesterol (TC), low-density lipoprotein cholesterol (LDL), high-density lipoprotein cholesterol (HDL), and triglycerides (TG).” (please see first sentence of the Introduction)

Lines 40-41: I feel this sentence is missing a reference as this is quite a significant point and inherently forms a basis for your rationale.

RESPONSE: Thank you, reference is added.

Line 76-78: While this may be a valid comment within the specific population study, I am not convinced how this rationale transfers to a general population. I do, however, agree to the fact there is a significant need for more research to be carried out in females (both young and old); due to the plethora of exercise studies utilising males. However, the importance of PE should transfer to all individuals. As far as I am aware, you haven’t specifically targeted this population (i.e. young mothers) within the study recruitment? I would suggest rewording this statement to reflect the above points.

RESPONSE: Thank you for profound explanation. This part of the text is reworded accordingly and now reads: “The effectiveness of PE is particularly important for young adult women, since they are nowadays frequently employed but, at the same time, face greater home duties and parental responsibilities than males.” (please see highlighted text 5th paragraph of the Introduction)

Methods:

Participant information: Require a statement in regard to the declaration of Helsinki.

RESPONSE: Thank you. The statements is included and text now reads: “They were informed that they could leave the program at any time. The study was in accordance to Helsinki declaration, and was approved by the Ethical Board of the University of Split, Faculty of Kinesiology, Split, Croatia (EBO: 2141-6775-234)” (please see subsection Participants – last sentence)

Lines 130/131: Just place units within the brackets, do not require “in”

RESPONSE: Corrected, thank you.

In regard to the endurance exercise protocol, as far as I understand, you utilise the Conconi test to assess heart rate threshold and have individuals train at approximately 5bpm under this. However, I noticed that you state: “participants performed different forms of endurance-based training protocols (i.e., continuous, interval, Fartlek)”. I wondered how this training modality was chosen, i.e. did they complete one session of each a week – or alternatively did individuals choose their own session each time. In which case, is it possible that individuals completed only interval training, or only continuous? Further, what were the specifics of these training modalities – i.e. on interval training sessions, was this a set protocol for all individuals, x seconds/minutes on, y seconds/minutes rest?

and

Further, I wondered as to how you managed progression. You correctly reassess their threshold every 2 weeks and adjust their training. However, is the training adjusted via speed, incline, both – or again is this on an individual basis? I do notice that in the discussion you mention “This allowed us to determine the intensity level (using the treadmill belt speed and heart rate) at which participants would most effectively exercise”. However, this is required within the methods and an explanation provided.  

and

I appreciate the aim of the study was to keep the set up as ‘real life’ and applicable as possible and you do mention that training sessions were programmed and controlled according to individual needs of each participant. However, the lack of specific detail does raise the question of whether there is too much variability. I would advise, where possible to be more specific so that the reader can understand exactly what occurred and what training stimulus was provided.

RESPONSE: We must agree that training programs were not sufficiently described. Therefore, this part of the text was systematically rewritten, and now reads: “During the endurance exercise, participants performed different forms of endurance-based training protocols (i.e., continuous, interval [starting from 2’ exercise + 1’ rest in first week up to 4’ exercise + 1’ rest last week], Fartlek), but heart rate values were constantly kept at least 5 bpm below the identified threshold. In general, participants were instructed to perform one form of endurance-based training protocol throughout the one week, and then to use another one (i.e. continuous training first week, Fartlek next week, and so on), but this was not mandatory, and participants frequently changed training forms within the single week (one session – one form of exercise). Generally, all participants participated in different form of endurance exercise, meaning that there was no chance that some participants performed only one form of exercise. Training sessions were programmed and controlled according to individual needs of each participant, and intensity was modified by carpet inclination and/or speed.” (Please see 2nd part of the 2nd paragraph of the subsection Physical exercise program). Thank you.

The resistance-based circuit training includes a large amount of exercises, targeting the whole body which is pleasing to see. While I appreciate you cannot always dictate the number of reps on each exercise (due to the nature of the modality), I did wonder whether the intensity of training was set for those which used machine or free weights. Naturally this would still be different between individuals but ensure training was relative between them. Similarly to my previous comments, perhaps a little more clarity on the specifics of the training would help the reader. 

RESPONSE: We must agree that resistance training was not sufficiently explained, especially with regard to details on testing. WE have tried to add more necessary details. text now reads: “Resistance training was organized and performed as circuit weight training using handheld weights, weight machines (Technogym, Cesena, Italy), and participants’ own body weight. Prior to study and initial testing, each participant performed two training sessions in order to accommodate herself with the training equipment and to familiarize with proper techniques and execution of exercise. Also, throughout these “familiarization sessions”, training instructor noted the weights that were identified to be appropriate for each exercise and each participant. It allowed us to make the chart where weights (resistances) were evidenced and later used in the resistance training.” (Please see 3rd paragraph of the “Physical exercise program” subsection. Thank you!)

Results:

Table 2: If post hoc statistics have been completed on the pre-post data, then I would suggest highlighting the significant differences within this table. It will very quickly allow the ready to see exactly what has changed with each training modality.

RESPONSE: Amended accordingly. Significant pre-post differences were noted with “*” in Table 2 for each group. Thank you.

Lines 227-234: Perhaps these comparisons could be displayed in a simpler way (similar to that in lines 219-222), e.g. significant differences were found between both training modalities and control for triceps skinfold, BF%, TC, HDL, LDL, TG and PG (all greater in control).

RESPONSE: Text is simplified as suggested and now reads: “When post-hoc analyses were calculated for post-testing results, significant between-group differences were found for: triceps skinfold, BF%, HDL, TG, and PG (all larger in control-group than in both training groups); TC and LDL (both larger in control-group than in endurance-training).” (Please see Results section).

In regard to the calorie intake, it is naturally an important result that you do not see any differences between groups nor between time points (in any group). However, I wondered if it may also be helpful to observe the macronutrient distribution. Whilst the calorie intake may be the same, it is naturally possible that individuals taking part in exercise 3 days a week may be inclined to substitute unhealthy (i.e. higher fat) foods for healthier alternatives. Further, this may be of particular interest in consideration of the control group reducing their TG despite not partaking in any exercise or altering their current activity. 

RESPONSE: Thank you for your support. According to your suggestion in the revised version of the manuscript, we included caloric intake for specific nutrients (Proteins, Carbohydrates, and Fats). It is now presented in the (new) Figure 2.

Discussion/conclusions:

Overall, a nicely constructed discussion with justified conclusions reached. You provide good context for the results you have found and discuss any anomalies between studies. I also believe you have adequately tackled the limitations of the study design and provide further rationale for enabling self-choice for participants.

RESPONSE: Thank you, we truly appreciate your reading, comments and expertise.

Reviewer 2 Report

The entire paper is extremely clear and very well written. I only have some minor comments and suggestions to the authors.

Introduction

Although the effectiveness of PE in women has been introduced (line 76), I think the authors should try to better explain this aspect when introducing the aim of the study. In fact, when reading that part (lines 93-97) it seems that this study has been carried out only because there is not enough research. However, giving more information about this when leading to the purpose of the study will add content on the whole study because at the moment the reader has the feeling that this study has been carried out because there are only few available while the reasons behind should be better included. Maybe the paper Aleksovska et al 2019 (doi: 10.1186/s40798-018-0173-9) could be of help.

Methods

Line 155: I don’t think there is a need to specify this “Specifics of each PE program are presented in the following text.”

What was the reason behind selecting 8 weeks for the duration of the intervention?

What about the duration of training session? How did you match endurance and resistance session for example in terms of energy expenditure?

What about the resistance training intensity? Did you have RM data?

Results

Table 2 needs some formatting as decimal numbers shifted lines.

All the results from the statistical analysis have been reported in Table 3, thus, to avoid redundancy, I will suggest the authors to not report F, p and ES values in the text (lines 204-218).

Why did the authors not include results about anaerobic testing and resistance testing?

Lines 219-235: Post-hoc results should be better presented.

Discussion

Line 243: Isn’t the sample to young to be indicated as premenopausal? I suggest delete the term in brackets.

Line 355: I agree with authors that “Self-motivation for physical exercise is among the most important factors of consistency in PE”.  To support this aspect, and maybe adding information also in the introduction or methods to explain your choice, I suggest reading the papers Condello et al 2017 (doi: 10.1186/s12966-017-0510-2) and Cortis et al 2017 (doi: 10.1371/journal.pone.0182709)

Author Response

The entire paper is extremely clear and very well written. I only have some minor comments and suggestions to the authors.

RESPONSE: Thank you! We followed your comments and amended the manuscript accordingly.

Introduction

Although the effectiveness of PE in women has been introduced (line 76), I think the authors should try to better explain this aspect when introducing the aim of the study. In fact, when reading that part (lines 93-97) it seems that this study has been carried out only because there is not enough research. However, giving more information about this when leading to the purpose of the study will add content on the whole study because at the moment the reader has the feeling that this study has been carried out because there are only few available while the reasons behind should be better included. Maybe the paper Aleksovska et al 2019 (doi: 10.1186/s40798-018-0173-9) could be of help.

RESPONSE: Thank you for your suggestion. We gladly accepted your suggestion, and included the findings of Aleksovska et al in our introduction. Text reads: “This is particularly important problem in women, and even systematic reviews confirmed that being female was negatively associated to PA and PE in all age categories (e.g. children, adolescents, and adults) [22] Specifically, it has been hypothesized that discouraging family/social environments could be a significant factor preventing, which is supportive to EU survey which indicates consistent decrease in PE in women, particularly those older than 24 years [22, 23]. Indeed, women are nowadays frequently employed but, at the same time, face greater home duties and parental responsibilities than males. This altogether limits their possibility to achieve regularity of PE, even if they are well informed about the necessity and importance of PE. As a result, information about the effectiveness of different PE types on indices of overall health in adult women is of high importance.” (Please see 5th paragraph of the Introduction)

Also, we believe that changes done in the Introduction following your later comments (please see last of your comments with regard to self-motivation) additionally strengthened the study background and explained its necessity.

Methods

Line 155: I don’t think there is a need to specify this “Specifics of each PE program are presented in the following text.”

RESPONSE: Amended accordingly. Thank you.

What was the reason behind selecting 8 weeks for the duration of the intervention?

RESPONSE: Thank you for noticing it. It is now explained and text reads: “Both PE programs lasted two months (8 weeks), with a training frequency of 3 sessions per week (Monday, Wednesday, Friday). Initially, authors discussed the duration of the experiment, and agreed that majority of females participated in 2-month (8-week) training, and after that moment significant drop-put occur. While authors were interested to investigate the effects of real-life training, the duration of 8-weeks was identified as being appropriate for the purpose of this study.” (please see 1st paragraph of the subsection “Physical exercise programs”).

What about the duration of training session? How did you match endurance and resistance session for example in terms of energy expenditure?

RESPONSE: In this study we did not match the trainings in terms of energy expenditure, but in term of duration. Therefore, both endurance- and resistance-training lasted: 45 min (first two weeks), 50 min (3rd and 4th week), etc. It is now clearly stated in the manuscript (please see end of the 1st paragraph of the subsection “Physical exercise programs).

What about the resistance training intensity? Did you have RM data?

RESPONSE: Actually, all participants were tested on their initial strength capacities and then again every two weeks. It is now specified and text reads: “Resistance training was organized and performed as circuit weight training using handheld weights, weight machines (Technogym, Cesena, Italy), and participants’ own body weight. Prior to study, each participant performed two training sessions in order to accommodate herself with the training equipment and to familiarize with proper techniques and execution of exercise. Also, throughout these “familiarization sessions”, training instructor noted the weights that were identified to be appropriate for each exercise and each participant. It allowed us to make the chart where weights (resistances) were evidenced and later used in the resistance training. Every two weeks (3 times over the study course) participants were invited to participate in additional individual training session, where first author of the study tested them on exercises used in circuit-weight-training in order to re-define individual training loads for the upcoming period of two weeks.” (Please see text on resistance training; subsection “Physical exercise program”).

Results

Table 2 needs some formatting as decimal numbers shifted lines.

RESPONSE: Formatted accordingly.

All the results from the statistical analysis have been reported in Table 3, thus, to avoid redundancy, I will suggest the authors to not report F, p and ES values in the text (lines 204-218).

RESPONSE: Amended accordingly. Text now reads: “Significant ANOVA main effects for “Group” were evidenced for thigh skinfold (medium ES), BF% (medium ES), and HDL (medium ES). Main effects for “Measurement” were significant for all studied variables, namely: body mass (large ES), BMI (large ES), triceps skinfold (large ES), thigh skinfold (large ES), suprailiac skinfold (large ES), BF% (large ES), TC (large ES), HDL (large ES), LDL (large ES), TG (large ES), and PG (large ES) (Table 3). The “Measurement x Group” interaction effects reached statistical significance for all studied anthropometric/body composition- and lipid-panel-variables, namely: body mass (large ES), BMI (large ES), triceps skinfold (large ES), thigh skinfold (large ES), suprailiac skinfold (large ES), BF% (large ES), TC (large ES), HDL (large ES), LDL (large ES), TG (large ES), and PG (large ES) (Table 3).” (Please see Results section 2nd and 3rd paragraph).

Why did the authors not include results about anaerobic testing and resistance testing?

RESPONSE: Indeed, this information will be interesting, but it should be reported individually for each participant. Therefore, in the revised version of the manuscript we stated that all individual results are available upon request. (Please see 2nd and 3rd para of the “Physical exercise program” subsection). Thank you.

Lines 219-235: Post-hoc results should be better presented.

RESPONSE: Post-hoc results are now systematically rewritten and text reads: “Post-hoc Scheffe test revealed significant within-group differences (e.g. pre- to post-testing differences) for both training-groups in body mass, BMI, triceps skinfold, thigh skinfold, subscapular skinfold, BF%, TC, HDL, LDL, PG, and TG (significant decrease in numerical results for all variables in training-groups). For Controls, significant post-hoc within-group differences were found for TG (significant decrease of TG over the course of the study). Between-group post-hoc analysis showed higher BMI for resistance- than for endurance-training-group in pre-testing results. When post-hoc analyses were calculated for post-testing results, significant between-group differences were found for: triceps skinfold, BF%, HDL, TG, and PG (all larger in control-group than in both training groups); TC and LDL (both larger in control-group than in endurance-training). No significant post-hoc differences were found between resistance-training and endurance-training group in post-testing.” (Please see Results section). Also, significance of pre- post-testing results are presented in Table2 as suggested by 1st Reviewer.

Discussion

Line 243: Isn’t the sample to young to be indicated as premenopausal? I suggest delete the term in brackets.

RESPONSE: Amended as suggested

Line 355: I agree with authors that “Self-motivation for physical exercise is among the most important factors of consistency in PE”.  To support this aspect, and maybe adding information also in the introduction or methods to explain your choice, I suggest reading the papers Condello et al 2017 (doi: 10.1186/s12966-017-0510-2) and Cortis et al 2017 (doi: 10.1371/journal.pone.0182709)

RESPONSE: Thank you for your suggestion and support. After reading suggested literature we included in the revised version of the paper. The references are now used in the Introduction, and text reads: “From the literature overview it is clear that a limited number of studies have examined and compared the effects of resistance training and endurance training in young adult women, especially with regard to important health-related indices such as the lipid panel and anthropometric/body composition indices. Next, information about the effectiveness of resistance training and endurance training in women who deliberately participate in certain PE programs is particularly scarce. Specifically, the initiation and consistency in PE is strongly correlated to different psychological and behavioral characteristics (including self-motivation, personal characteristics, environmental specifics), and it is hard to expect that individuals will partake in exercise program they don’t prefer [29, 30]. Consequently, this study aimed to evaluate, etc.” (Please see last paragraph of the Introduction.

Thank you once again for your valuable and constructive comments.

Staying at your disposal!

Authors

Reviewer 3 Report

The authors need to justify why that study was done with participants of which most seemed to be unproblematic with respect to lipid profile, body composition and BMI. What percentage of the participants was out of the normal range values for lipid profile, what percentage did not have normal weight etc. What was the rationale for the study with apparently healthy participants.

In the discussion, the authors need to provide, if possible, a comparison whether their observations are comparable with other training studies with healthy adult women with more or less similar baseline characteristics. There are now statements on comparison with obese women and older women.

Changes in plasma glucose need to be discussed.

Ls 17-18 and 107-108. I suggest to change “23.11 ± 3.07 years; initial body height: 165.22 ± 5.83 17 cm; body mass: 66.79 ± 7.23 kg; BMI: 24.37 ± 2.57 kg/m2)” to “23 ± 3 years; initial body height: 165 ± 6 cm; body mass: 66.8 ± 7.2 kg; BMI: 24.37 ± 2.57 kg/m2)”

L22. Plasma glucose is not a lipid. Please correct.

Ls 29-31. The final statement of the abstract needs to be justified by preceding abstract information. Please revise the abstract.

L42. Change “gender, environment” to “gender, and environment”

L59. Change “transportation, occupational” to “transportation, and occupational”

L62. Chang “Medicine, Centers” to “Medicine, and the Centers”

Ls 76-81. Information here is speculative and when these statements are important the study should have recruited employed mothers but that was not the case. I suggest to revise and even delete a lot of this info and keep the focus on the type of training. Please revise “The effectiveness of PE is particularly important for young adult women, especially if they are mothers. Women are frequently employed but, at the same time, face greater home duties and maternal responsibilities than males. This altogether limits the possibility to achieve regularity of PE among adult women, even if they are well informed about the necessity and importance of PE. As a result, information about the effectiveness of different PE types on indices of overall health in adult women is of high importance.”

L154. Just mention between 16.00 and 21.00 or change to late afternoon and early evening.

Ls 169-172. Detailed information on the resistance training is provided. As it seems that heart rate has been recorded for the endurance training sessions, I suggest to provide what the heart rate values were for the training sessions. In addition, it is difficult to see how with Fartlek allowed that heart rates were below 5 bpm below a certain threshold. Also, in the discussion is mention of walking uphill but no mention of that in the methods section. More detailed information on the endurance training sessions needs to be provided.

Table 2. Body mass changed in the training groups by about 5%. Is that what can be normally accepted, as it is almost 0.5 kg/week with no dietary restrictions. This observation needs discussion and interpretation.

Ls 251-256. There are comparisons with other studies with women as participants but I suggest to focus only on training studies with females with more or less similar baseline characteristics as your study. Some of the studies you refer to are on obese women and middle aged and older women. That is not a fair comparison. Please revise this section.

Ls 298-301. Comparison of observations of resistance training studies with postmenopausal women and obese women is not appropriate considering your subject characteristics. Please make only comparison with studies that have females with more or less similar baseline characteristics. Please revise.

Ls 306-328. The authors need to discuss whether a change of the lipid profile or body composition is needed when it is not considered problematic. What are normative values for the components of the lipid profile and what percentage of the participants were outside of accepted normal range values. It seems also that most participants had normal BMI so what is the point of changing that and still having normal BMI.

Author Response

The authors need to justify why that study was done with participants of which most seemed to be unproblematic with respect to lipid profile, body composition and BMI. What percentage of the participants was out of the normal range values for lipid profile, what percentage did not have normal weight etc. What was the rationale for the study with apparently healthy participants.

RESPONSE: Thank you for your suggestion. Indeed, the rationale of the study was not properly described in the original version of the paper. In this revision we tried to elaborate it more specifically. The text reads: “This is particularly important problem in women, and even systematic reviews confirmed that being female was negatively associated to PA and PE in all age categories (e.g. children, adolescents, and adults) [22] Specifically, it has been hypothesized that discouraging family/social environments could be a significant factor preventing, which is supportive to EU survey which indicates consistent decrease in PE in women, particularly those older than 24 years [22, 23]. Indeed, women are nowadays frequently employed but, at the same time, face greater home duties and parental responsibilities than males. This altogether limits their possibility to achieve regularity of PE, even if they are well informed about the necessity and importance of PE. As a result, information about the effectiveness of different PE types on indices of overall health in adult women is of high importance.” (please see highlighted text – 5th paragraph pf the Introduction). With regard to “values for lipid profile”, it is now more specifically explained in Participants subsection. Text reads: “On the basis of pre-testing lipid-profile results (please see later for details on testing), the above optimal level of TC was evidenced in 25% of participants (5% with high levels of TC), 23% of participants had above optimal level of LDL (6% with high levels of LDL), and <10% of participants had above optimal levels of TC [31].” Please see subsection on Participants

In the discussion, the authors need to provide, if possible, a comparison whether their observations are comparable with other training studies with healthy adult women with more or less similar baseline characteristics. There are now statements on comparison with obese women and older women.

RESPONSE: Thank you for your suggestion. The results of our study are now compared with other studies with health subjects. Text reads. “Also, in recent study with healthy young females (25-30 years of age), Kyrolainen et al. evidenced changes in lipid panel indicators, as a result of 9-week endurance training (Kyrolainen, Hackney et al. 2018). More specifically, TG changes were not significant (1.17±0.34 and 1.01±0.29 mmol/L), TC was significantly reduced (5.0±.05 mmol/L and 4.4±0.7 mmol/L), while HDL levels significantly increased (1.37±0.29 and 1.51±0.27 mmol/L, for pre-testing and after training-period, respectively) (Kyrolainen, Hackney et al. 2018). Somewhat better results in our study may be explained by the fact that our participants performed treadmill-endurance training, while our respected colleagues applied indoor cycling as a type of endurance training. Knowing the differences in overall energy expenditure between these two forms of exercising (i.e. treadmill walking/running engage larger percentage of the musculature than cycling), the somewhat better effects in our study are understandable (Millet, Vleck et al. 2009). However, despite some differences, our results are supportive to previous findings where positive effects of exercise are evidenced in healthy young females.” (Please see new 3rd paragraph of the subsection 4.2. Positive effects of endurance and resistance training on lipid profile. Thank you!)

Changes in plasma glucose need to be discussed.

RESPONSE: Thank you. In the revised version of the manuscript, the first paragraph of the subsection 4.2 discusses the changes in PG. Text reads: “The significant reduction of the PG levels in our participants was expected, simply because carbohydrates (e.g. PG and muscle glycogen) are primary energy source during and kind of PA (Holloszy and Kohrt 1996). On the other hand, the differential effects of studied PE-programs were also possible. In brief, the PG utilization increases with the intensity of PE, simply because of the increase in glucose utilization by muscles. Throughout the PE, the utilization is affected (e.g. increased) by: (i) intensity of the activation of each muscle unit, (ii) increase in the number of active muscle units, and/or (iii) both (Coggan 1991). Although we can ignore the fact that both of these mechanisms are more evidenced in resistance training (i.e. resistance training increase the intensity and number of activated muscle unites to a greater extent than endurance-training), plasma glucose utilization also increases with the duration of exercise. It therefore may at least partially compensate the theoretically expected superior effects of resistance-training even in our study.”

Ls 17-18 and 107-108. I suggest to change “23.11 ± 3.07 years; initial body height: 165.22 ± 5.83 17 cm; body mass: 66.79 ± 7.23 kg; BMI: 24.37 ± 2.57 kg/m2)” to “23 ± 3 years; initial body height: 165 ± 6 cm; body mass: 66.8 ± 7.2 kg; BMI: 24.37 ± 2.57 kg/m2)”

RESPONSE: Amended accordingly.

L22. Plasma glucose is not a lipid. Please correct.

RESPONSE: Corrected. Thank you for noticing it.

Ls 29-31. The final statement of the abstract needs to be justified by preceding abstract information. Please revise the abstract.

RESPONSE: Thank you. Abstract is revised accordingly. In brief, since in the Conclusion we “target” pre-study screening of the fitness status, this issue is highlighted in the Methods, and text reads: “Methods: Participants were women (n = 57; age: 23 ± 3 years; initial body height: 165 ± 6 cm; body mass: 66.79 ± 7.23 kg; BMI: 24.37 ± 2.57 kg/m2) divided into the ET group (n = 20), RT group (n = 19), and non-exercising control group (n = 18). All participants were tested for cardiovascular risk factors (CRF) including: total cholesterol, HDL, LDL, triglycerides, glucose, and anthropometric/body composition (body mass, BMI, skinfold measures, body fat %) at the beginning and at the end of the study. Over the 8 weeks, the ET group trained 3 times/week on a treadmill, while the RT group participated in equal number of circuit weight training sessions. Both types of training were planned according to participants’ pre-study fitness status.”

L42. Change “gender, environment” to “gender, and environment”

RESPONSE: Amended accordingly.

L59. Change “transportation, occupational” to “transportation, and occupational”

RESPONSE: Amended accordingly.

L62. Chang “Medicine, Centers” to “Medicine, and the Centers”

RESPONSE: Amended accordingly.

Ls 76-81. Information here is speculative and when these statements are important the study should have recruited employed mothers but that was not the case. I suggest to revise and even delete a lot of this info and keep the focus on the type of training. Please revise “The effectiveness of PE is particularly important for young adult women, especially if they are mothers. Women are frequently employed but, at the same time, face greater home duties and maternal responsibilities than males. This altogether limits the possibility to achieve regularity of PE among adult women, even if they are well informed about the necessity and importance of PE. As a result, information about the effectiveness of different PE types on indices of overall health in adult women is of high importance.”

RESPONSE: Thank you. Indeed, the similar suggestion was provided by other Reviewers. This part of the text is systematically rewritten, references are added and text now reads: “Although the benefits of PA in everyday life are relatively well documented, the problem of regularity in PA is crucial. Modern life offers many advantages, but many young adults are facing the problems of extremely busy schedules, including “overload” and or “increased stress levels” (APA. Annual Convention report. September 2013 , Duxbury 2003). Therefore, it is of high importance to find the most appropriate type of PE for the various needs of different subjects. This is particularly important problem in women, and even systematic reviews confirmed that being female was negatively associated to PA and PE in all age categories (e.g. children, adolescents, and adults) (Aleksovska, Puggina et al. 2019) Specifically, it has been hypothesized that discouraging family/social environments could be a significant factor preventing, which is supportive to EU survey which indicates consistent decrease in PE in women, particularly those older than 24 years (Special and 2014;412:1–135. , Aleksovska, Puggina et al. 2019). Indeed, women are nowadays frequently employed but, at the same time, face greater home duties and parental responsibilities than males. This altogether limits their possibility to achieve regularity of PE, even if they are well informed about the necessity and importance of PE. As a result, information about the effectiveness of different PE types on indices of overall health in adult women is of high importance.” (please see highlighted text in the 5th paragraph of the Introduction. Thank you.

L154. Just mention between 16.00 and 21.00 or change to late afternoon and early evening.

RESPONSE: Thank you, we retained hours.

Ls 169-172. Detailed information on the resistance training is provided. As it seems that heart rate has been recorded for the endurance training sessions, I suggest to provide what the heart rate values were for the training sessions. In addition, it is difficult to see how with Fartlek allowed that heart rates were below 5 bpm below a certain threshold. Also, in the discussion is mention of walking uphill but no mention of that in the methods section. More detailed information on the endurance training sessions needs to be provided.

REPONSE: In this version of the manuscript more details are provided about both types of training. For example, the text about endurance training now contains (in addition to original text): “During the endurance exercise, participants performed different forms of endurance-based training protocols (i.e., continuous, interval [starting from 2’ exercise + 1’ rest in first week up to 4’ exercise + 1’ rest last week], Fartlek), but heart rate values were constantly kept at least 5 bpm below the identified threshold. In general, participants were instructed to perform one form of endurance-based training protocol throughout the one week, and then to use another one (i.e. continuous training first week, Fartlek next week, and so on), but this was not mandatory, and participants frequently changed training forms within the single week. Generally, all participants participated in different forms of endurance exercise, meaning that there none of the participants performed single form of exercise (i.e. continuous, interval). Training sessions were programmed and controlled according to individual needs of each participant, and intensity was modified by inclination and/or speed of the treadmill-carpet.” (Please see highlighted text in 2nd part of the subheading 3.3. Physical exercise programs. Thank you.)

Table 2. Body mass changed in the training groups by about 5%. Is that what can be normally accepted, as it is almost 0.5 kg/week with no dietary restrictions. This observation needs discussion and interpretation.

RESPONSE: Actually, this is in accordance with previous reports, and participants initial status with regard to BMI and BF values (i.e. with food restriction the changes of <1kg are accepted). However, we briefly discussed it in the revised version of the manuscript and text reads: “Finally, Lithuanian study where authors examined effects of two-month endurance exercise program performed by indoor cycling, induced significant reduction of body mass of 1kg in young health females (in our study 2 kg), but in this study authors examined participants with initially lower BMI and BF% than we have observed herein (BMI: 23 and 24 kg/m2, BF: 31 and 35% in Lithuanian and our study, respectively), which naturally resulted even in relatively smaller numerical changes in body mass (Stasiulis, Mockiene et al. 2010).” (Please see highlighted text in 1st para of the subsection 4.1. Positive effects on anthropometrics. Thank you.)

Ls 251-256. There are comparisons with other studies with women as participants but I suggest to focus only on training studies with females with more or less similar baseline characteristics as your study. Some of the studies you refer to are on obese women and middle aged and older women. That is not a fair comparison. Please revise this section.

RESPONSE: Indeed, most of the comparisons were done with studies involving obese women, but we didn’t want to be “unfair”, but simply, most of the studies that examined the problem observed such participants. It is now corrected, and text is amended accordingly, so we retained and discussed only studies who involved healthy, relatively young female participants. Text reads: “Endurance training induced significant, positive effects on anthropometric/body composition indices. Such results were expected, and similar improvements were reported in other studies where different forms of endurance (aerobic) training were studied in women. For example, a recent Spanish study with sedentary females investigated the effects of a 16-week Zumba aerobic dance program and revealed positive effects on BMI and body fat percentage (Barranco-Ruiz, Ramirez-Velez et al. 2019). The Lithuanian study where authors examined effects of two-month endurance exercise program performed by indoor cycling, induced significant reduction of body mass of 1kg in young health females (in our study 2 kg), but in this study authors examined participants with initially lower BMI and BF% than we have observed herein (BMI: 23 and 24 kg/m2, BF: 31 and 35% in Lithuanian and our study, respectively), which naturally resulted even in relatively smaller numerical changes in body mass (Stasiulis, Mockiene et al. 2010). Positive results of endurance training on different indices of body build have been reported in other studies with young health females as well (LeMura, von Duvillard et al. 2000). On the other hand, it is important to note that the duration of our study was evidently shorter (i.e., 8 weeks) than the durations for the majority of previous studies, which regularly lasted 3 months or more (LeMura, von Duvillard et al. 2000, Barranco-Ruiz, Ramirez-Velez et al. 2019). In explaining the significant effects on body composition indices in our study, irrespective of relatively short duration, two most important issues should be highlighted and discussed: (i) controlled intensity and (ii) type of endurance exercise.” (Please see 1sta paragraph of the subsection 4.1)

 Ls 298-301. Comparison of observations of resistance training studies with postmenopausal women and obese women is not appropriate considering your subject characteristics. Please make only comparison with studies that have females with more or less similar baseline characteristics. Please revise.

RESPONSE: Text is revised accordingly, and only results of studies done with similar samples are presented. It reads: “Our results showed positive effects of aerobic endurance exercise on changes in lipid profile, and this is generally consistent with previous reports where authors examined longer training programs. For example, 6-month study of Dunn et al., who examined a mixed sample of healthy, sedentary men and women, showed promising results, with significantly positive changes in total cholesterol, LDL, and the total/HDL ratio (Dunn, Marcus et al. 1997). One of the rare studies exclusively examining females evaluated the effects of 16 weeks of endurance exercise, and a significant increase in HDL and decrease in the concentration of triglycerides were reported (LeMura, von Duvillard et al. 2000). In recent study with healthy young females (25-30 years of age), Kyrolainen et al. evidenced changes in lipid panel indicators, as a result of 9-week endurance training (Kyrolainen, Hackney et al. 2018). More specifically, TG changes were not significant (1.17±0.34 and 1.01±0.29 mmol/L), TC was significantly reduced (5.0±.05 mmol/L and 4.4±0.7 mmol/L), while HDL levels significantly increased (1.37±0.29 and 1.51±0.27 mmol/L, before and after study period, respectively) (Kyrolainen, Hackney et al. 2018). Somewhat better imporvements in our study may be explained by the fact that our participants performed treadmill-endurance training, while our respected colleagues applied indoor cycling as a type of endurance training. Knowing the differences in overall energy expenditure between these two forms of exercising (i.e. treadmill walking/running engage larger percentage of the musculature than cycling), the somewhat better effects in our study are understandable (Millet, Vleck et al. 2009). However, despite of some differences, our results are supportive to previous findings where positive effects of exercise are evidenced in healthy young females.” (please see 2nd paragraph of the subsection 4.2)

 Ls 306-328. The authors need to discuss whether a change of the lipid profile or body composition is needed when it is not considered problematic. What are normative values for the components of the lipid profile and what percentage of the participants were outside of accepted normal range values. It seems also that most participants had normal BMI so what is the point of changing that and still having normal BMI.

RESPONSE: Thank you for noticing it. We tried to discuss the issue of “training effects” in apparently healthy participants. This is done in Conclusion section. Text reads: “This study involved apparently healthy female participants with 32% overweight/obese, and 25% of participants who had above-optimal levels of lipid-panel indicators. Therefore, the necessity of the study may be questioned. However, because of the alarming trends of increase in cardiovascular-risks, it is important to determine the eventual positive effects of PE even in low-risk populations and groups. Namely, this study has demonstrated that exposure to individually preferred PE can induce significant health-benefits, and can therefore prevent occurrence of various cardiovascular risks. The information of such kind can be used in targeted and preventive public health efforts.” (please see highlighted text in Conclusion)

Staying at your disposal.

Authors

Round 2

Reviewer 3 Report

L:184. I suggest to delete “and authors are…… request”.

L190. The authors acknowledge the importance of exercise intensity (i.e. L 293). Therefore, please provide the heart rate values of the identified threshold. And based on age, what percentage were those of age-predicted maximum heart rate. This information would inform readers on the exercise intensity for the endurance training sessions.

L203. I suggest to delete “(authors will gladly provide details upon request)”.

L284. “(in our study 2 kg),” This is not correct, please change.

L338. Change “plassma”.

Ls 339-340. Please revise “The significant reduction of the PG levels in our participants was expected, simply because carbohydrates (e.g. PG and muscle glycogen) are primary energy source during and kind of PA [54].” This statement is incorrect. Regarding the reason for lower PG, what about changes in insulin sensitivity? The authors need to interpret the lower PG levels.

L372. Change “imporvements”

L373. Please revise “while our respected colleagues”. This is not common in scientific writing”

Author Response

Dear Madam/Sir

Thank you again for your detailed comments. We amended the manuscript accordingly. Please see bellow for responses. Authors

L:184. I suggest to delete “and authors are…… request”.

RESPONSE: Amended as suggested.

L190. The authors acknowledge the importance of exercise intensity (i.e. L 293). Therefore, please provide the heart rate values of the identified threshold. And based on age, what percentage were those of age-predicted maximum heart rate. This information would inform readers on the exercise intensity for the endurance training sessions.

RESPONSE: Details are added, and text now reads: "Participants exercised in a range 5-to-30 beats bellow anaerobic threshold, which corresponds to 120-155 beats/min (60-80% of maximal heart rate), in average. "

L203. I suggest to delete “(authors will gladly provide details upon request)”.

RESPONSE: Amended accordingly.

L284. “(in our study 2 kg),” This is not correct, please change.

RESPONSE: Corrected, thank you.

L338. Change “plassma”.

RESPONSE: Corrected

Ls 339-340. Please revise “The significant reduction of the PG levels in our participants was expected, simply because carbohydrates (e.g. PG and muscle glycogen) are primary energy source during and kind of PA [54].” This statement is incorrect. Regarding the reason for lower PG, what about changes in insulin sensitivity? The authors need to interpret the lower PG levels.

RESPONSE: Revised accordingly. Text reads: "The significant reduction of the PG levels in our participants was expected, because: (i) carbohydrates (e.g. PG and muscle glycogen) are one of the main energy sources during PA, while (ii) PA increases insulin sensitivity, which directly reduces the PG concentration in trained subjects [54-56]. "

L372. Change “imporvements”

RESPONSE: Corrected

L373. Please revise “while our respected colleagues”. This is not common in scientific writing”

RESPONSE: Revised (text reads: "while Kyrolainen et al..."